# Oxidative Stress and Inflammation Caused by Cisplatin Ototoxicity

**DOI:** 10.3390/antiox10121919

**Published:** 2021-11-29

**Authors:** Vickram Ramkumar, Debashree Mukherjea, Asmita Dhukhwa, Leonard P. Rybak

**Affiliations:** 1Department of Pharmacology, School of Medicine, Southern Illinois University, 801 N. Rutledge Street, Springfield, IL 62702, USA; vramkumar@siumed.edu (V.R.); adhukhwa59@siumed.edu (A.D.); 2Department of Otolaryngology, School of Medicine, Southern Illinois University, 801 N. Rutledge Street, Springfield, IL 62702, USA; dmukherjea@siumed.edu

**Keywords:** oxidative stress, cisplatin, inflammation, heat shock proteins, G-protein coupled receptors

## Abstract

Hearing loss is a significant health problem that can result from a variety of exogenous insults that generate oxidative stress and inflammation. This can produce cellular damage and impairment of hearing. Radiation damage, ageing, damage produced by cochlear implantation, acoustic trauma and ototoxic drug exposure can all generate reactive oxygen species in the inner ear with loss of sensory cells and hearing loss. Cisplatin ototoxicity is one of the major causes of hearing loss in children and adults. This review will address cisplatin ototoxicity. It includes discussion of the mechanisms associated with cisplatin-induced hearing loss including uptake pathways for cisplatin entry, oxidative stress due to overpowering antioxidant defense mechanisms, and the recently described toxic pathways that are activated by cisplatin, including necroptosis and ferroptosis. The cochlea contains G-protein coupled receptors that can be activated to provide protection. These include adenosine A1 receptors, cannabinoid 2 receptors (CB2) and the Sphingosine 1-Phosphate Receptor 2 (S1PR2). A variety of heat shock proteins (HSPs) can be up-regulated in the cochlea. The use of exosomes offers a novel method of delivery of HSPs to provide protection. A reversible MET channel blocker that can be administered orally may block cisplatin uptake into the cochlear cells. Several protective agents in preclinical studies have been shown to not interfere with cisplatin efficacy. Statins have shown efficacy in reducing cisplatin ototoxicity without compromising patient response to treatment. Additional clinical trials could provide exciting findings in the prevention of cisplatin ototoxicity.

## 1. Introduction

The cochlea can be damaged by a variety of insults that can result in acquired sensorineural hearing loss. Hearing loss can result from ototoxic drugs, noise trauma, and injury from cochlear implant insertion, ageing and radiation. A common theme demonstrated by these harmful entities is the production of oxidative stress by free radical generation leading to inflammation and loss of sensory cells and reduction in hearing acuity. This review will address cisplatin ototoxicity. It includes an up to date presentation of the mechanisms associated with cisplatin-induced hearing loss, current treatments to ameliorate cisplatin ototoxicity and novel and exciting future treatments that are being explored.

## 2. Cisplatin Ototoxicity

Cisplatin is widely used as a chemotherapeutic agent to treat solid malignant tumors in adults and children [1]. Unfortunately, permanent sensorineural hearing loss is a frequent complication of cisplatin therapy. The higher frequencies are affected initially, but subsequently hearing loss in the middle frequency range can appear when higher doses are administered. Children are more likely than adults to develop delayed ototoxicity from cisplatin [2]. Long-term retention of platinum in the cochlea after cisplatin chemotherapy has been demonstrated in pediatric patients [3]. Young children are more susceptible to cisplatin induced hearing loss. Hearing loss from cisplatin develops early during chemotherapy and its cumulative incidence is increased with total cumulative doses of cisplatin and the administration of other ototoxic drugs [4].

Animal models for cisplatin ototoxicity reveal that outer hair cells (OHCs) of the first row of basal turn of the cochlea are the targets for initial damage [3,5,6]. Injury extends to additional rows of OHCs of the basal turn, and then apically with additional dosing [5]. Progressive damage to OHCs include parietal membrane dilation, cuticular plate softening, vacuole formation and accumulation of lysosomes in the apical portion of the cells. Stereocilia fusion can be seen on OHCs and inner hair cells (IHCs) [6]. Platinum preferentially accumulates to the highest concentration in the stria vascularis [3] leading to strial damage, particularly with high dose cisplatin [7].

## 3. Mechanisms of Ototoxicity

The molecular mechanisms by which cisplatin causes cochlear damage and hearing loss are still being studied. Cisplatin needs to enter cells in the cochlea in order to cause cell death. Cisplatin may enter cells by passive diffusion and facilitated transport [8,9]. Cisplatin may also gain access to target cells in the cochlea by using transporters. These include copper transporter 1 (CTR1) [10] and/or organic cation transporters (OCTs), such as OCT 2 [11]. Upon entry into the cells cisplatin may be hydrolyzed by water to generate aqua-cisplatin complexes [12]. These complexes are highly toxic and can damage DNA. This can upregulate ataxia telangiectasia mutated (ATM). ATM can activate the tumor suppressor molecule p53 [13]; this increases the expression of the pro-apoptotic protein, Bcl-associated X (Bax). Bax increases mitochondrial membrane permeability leading to cytochrome c release from mitochondria through activation of caspase 3 [14]. 

Mechanoelectrical transduction (MET) channels may provide a route of entry for cisplatin into hair cells. Thus, partial or reversible blockade of MET channels may reduce cisplatin ototoxicity. Cisplatin has been found to block MET channels in chick cochlear hair cells in a dose and voltage dependent manner [15]. Iontophoresis of cisplatin into the scala media of guinea pigs resulted in hearing loss by blocking OHC MET channels [16]. Chemical inhibition of MET channels with quinine or EGTA blocked cisplatin uptake and prevented hair cell death in zebrafish [17]. A reversible, high-affinity permeant blocker of the MET channel, ORC-13661, protected neonatal mouse OHCs from cisplatin damage [18]. This finding suggests that less cisplatin was able to enter these OHCs in the presence of MET channel blockade, and thus they were protected from damage. Mice expressing fluorescently labeled transmembrane channel-like protein isoform (TMCs), which are associated with the MET channels exhibit a three-fold increase in the number of TMC1 molecules in the tip of each stereocilium from the apex to the base of the cochlea [19]. These findings could partially explain the greater sensitivity of the base of the cochlea to cisplatin damage resulting in larger threshold shifts at higher frequencies. Potential mechanistic targets underlying cisplatin ototoxicity are presented in Figure 1.

## 4. Oxidative Stress

The cochlea has a high rate of metabolism. It requires an active antioxidant defense system in order to manage reactive oxygen species (ROS) generation from normal metabolism to function properly to maintain healthy hearing. This defense mechanism can be overwhelmed when the cochlea is subjected to extreme oxidative stress from noise or ototoxic drugs. Cisplatin ototoxicity is strongly associated with increased oxidative stress in the cochlea [20,21]. A key target of cisplatin in the cochlea is the NOX3 isoform of NADPH oxidase [22]. This enzyme is uniquely expressed in the cochlea. Rats treated with trans-tympanic siRNA for NOX3 had reduced ototoxicity from cisplatin [23]. Furthermore, mice treated with cisplatin exhibited increased expression of NOX3 in (OHCs) and supporting cells (SCs) and significant high-frequency hearing loss with auditory brainstem response (ABR) testing. However, NOX3-knockout mice had significantly smaller ABR threshold shifts and fewer OHC loss in all segments of the basal turn of the cochlea. NOX3 expression was found to be increased following cisplatin exposure, aging and noise trauma. The greatest increase of NOX3 expression was found in cisplatin treated mice. Nox3 upregulation, particularly in the basal turn SCs and OHCs, appears to be directly involved in the development of sensorineural hearing loss (SNHL) [24]. Xanthine oxidase is another enzyme that could be a source of ROS generation in the cochlea [25]. Allopurinol is an inhibitor of this enzyme that has been shown to reduce cisplatin ototoxicity when administered in combination with ebselen, a glutathione peroxidase mimic [25]. 

Cisplatin can also activate nitric oxide synthases in the cochlea [26,27], thus increasing the level of nitric oxide (NO) in the cochlea. NO can react with proteins in the cochlea, resulting in their nitration and loss of function. NO can also react with O_2-_ produced by cisplatin upregulation of NOX3 to generate the highly reactive peroxynitrite free radical (ONOO-). Cisplatin induced nitrative stress in the cochlea leads to the nitration of LIM domain only 4 (LMO4), a transcriptional regulator, and decreases its cochlear expression levels. LMO4 is a transcriptional regulator that controls the choice between cell survival and cell death. Nitration of LMO4 appears to be related to cisplatin-induced ototoxicity [28]. Furthermore, LMO4 knockout animals are more susceptible to cisplatin induced hearing loss and apoptosis of cochlea cells [29]. Thus, both ROS and RNS can react with and damage proteins, lipids and DNA within the cell to inhibit their normal function leading to cellular damage and loss of hearing (see Figure 1).

Cisplatin ototoxicity has recently been associated with ferroptosis [30,31]. Ferroptosis is a type of cell death that is not apoptotic. It can be activated by various small molecules, such as cisplatin, or it can be induced by inhibition of the biosynthesis of glutathione or inhibition of glutathione peroxidase 4 (GPx4). It involves iron-dependent increase in lipid ROS and depletion of polyunsaturated fatty acids in plasma membranes [32]. Cisplatin was demonstrated to activate ferroptosis in HEI-OC1 cells, leading to increased lipid peroxidation, iron accumulation and reduction of the mitochondrial membrane potential (MMP). A specific inhibitor of ferroptosis, ferrostatin-1 was shown to diminish cell death due to cisplatin in HEI-OC1 cells and in mouse cochlear explants by improving mitochondrial function [30,31]. Blockade of autophagy using chloroquine was found to ameliorate ferroptosis induced by cisplatin in HEI-O C1 cells, suggesting that the cell death induced by ferroptosis was associated with upregulation of autophagy [33]. However, targeting ferroptosis to prevent cisplatin induced hearing loss could be problematic, since ferroptosis is one of the mechanisms that underlie cisplatin induced killing of tumor cells [34]. Systemic treatment with a ferroptosis inhibitor could interfere with cisplatin efficacy to treat cancer. This could be avoided by using localized intratympanic administration of a ferroptosis inhibitor.

Cisplatin ototoxicity has also been associated with necroptosis. Necroptosis is a novel form of cell necrosis that serves as an alternative cell death pathway. It differs from necrotic cell death, which is passive. Rather it follows a program in cells that is mediated by the receptor-interacting protein kinases (RIPKs), RIPK1 and RIPK3. The intraperitoneal administration of Necrostatin-1s, a specific inhibitor of RIPK1 [35], protected mice against cisplatin induced hearing loss. Genetic inhibition of RIPK3 also protected against cisplatin ototoxicity in mice [36]. 

Oxidative stress can deplete the levels of antioxidant enzymes. Excess ROS production can also cause mitochondria to release cytochrome c which results in apoptosis [14]. Cisplatin also reduces the levels of antioxidant enzymes, including superoxide dismutase (SOD), glutathione reductase (GR), glutathione S-transferase (GST) and glutathione peroxidase (GSH-Px) [37,38]. 

Transient receptor potential (TRP) channels play an important role in mediating sensations, such as taste, touch and smell. They are also associated with neurogenic pain and inflammation [39,40,41]. These are nonselective cation permeable channels that can be activated by environmental or endogenous stimuli [42] including toxic reactive aldehydes [43] generated by cisplatin-mediated lipid peroxidation resulting from ROS generation. Several types of TRP channels have been identified in the cochlea [41]. TRPV1 channels are activated by oxidative stress, mediated by NOX3. NOX3 is activated by cisplatin leading to the production of ROS and TRPV1 upregulation. Knockdown of *TRPV1* expression was achieved by transtympanic administration of *TRPV1* siRNA. This protected against cisplatin-induced hearing loss. This protection was accompanied by reduced calcium influx and *NOX3* expression, resulting in amelioration of ROS production, inflammation and apoptosis in the cochlea [40]. 

## 5. Endogenous Antioxidant Defense System of the Cochlea

Under normal conditions of auditory stimulation, ROS generated in the cochlea can be scavenged by an antioxidant defense system. This system consists of glutathione and enzymes that synthesize and regenerate glutathione and enzymes that detoxify ROS. ROS detoxification mechanisms are critical for protection of the cochlea since cisplatin produces excessive ROS in the cochlea that damage vulnerable cells by generating lipid peroxides, including malondialdehyde and 4-hydroxynonenal (4-HNE). 

The highest levels of glutathione are present in the basal and intermediate cells of the stria vascularis and in the spiral ligament fibrocytes of the guinea pig cochlea [44] and are associated with the content of glutathione S-transferases [45]. Glutathione reductase and glutathione peroxidase (GPx) regenerate glutathione from its oxidized form. Glutathione-S-transferases are involved in cell detoxification. Of these, the glutathione transferase (GST) detoxification system converts a toxic compound into a less toxic form by conjugating the toxin to SH groups of reduced glutathione [46]. One of these enzymes, glutathione transferase α4 (GSTA4), has been shown to reduce cisplatin ototoxicity by detoxifying the toxic byproduct, 4-hydroxynonenal (4-HNE) in the cochlea of female mice [47]. 

Additional antioxidant enzymes are expressed in the cochlea. These include superoxide dismutases (SOD) and catalase (CAT). The SOD isoforms expressed in the cochlea include a cytosolic Cu/Zn-SOD isoform and a mitochondrial Mn-regulated isoform (Mn-SOD) [48]. The Mn-SOD isoform is found in the metabolically active tissues in the cochlea, including the lateral wall tissues (stria vascularis, spiral ligament, spiral prominence, spiral limbus and the organ of Corti [49]. CAT and GPx can detoxify H_2_O_2_. SOD and CAT act sequentially to reduce superoxide (O_2_^−^) to O_2_ and H_2_O. Dismutation of O_2_^−^ by superoxide dismutase (SOD) produces relatively more reactive molecule, hydrogen peroxide (H_2_O_2_), which may be detoxified by CAT. However, if metal ions are present, the H_2_O_2_ may be converted into the more toxic and reactive hydroxyl radical (OH^−^). 

Additional antioxidant defense mechanisms are present in the cochlea. Vitamin C (ascorbic acid) is an endogenous antioxidant that has been shown to be present in cochlear fluid of guinea pigs in greater concentration than in plasma [50]. Cochlear oxidative stress can up-regulate kidney injury molecule-1 (KIM-1) which may help repair damaged cochlear tissue if it is not overwhelmed by cisplatin [51]. 

Heat shock proteins, including HSP-27, HSP32, HSP40, HSP60, HSP70 and HSP90 protects against cisplatin-induced hair cell death. These protective molecules not only inhibit apoptosis, but they also upregulate SOD [52,53]. 

Adult mouse utricles were cultured and subjected to heat shock. This treatment induced a strong up-regulation of *HSP70* mRNA and a smaller increase in the expression of *HSP90* and *HSP27*. The up-regulation of these HSPs resulted in protection of utricular cells against cisplatin induced cell death [53]. Another study subjected mouse utricular cultures to heat shock or transfection with adenovirus expressing *HSP70*. HSP70 was induced in glia-like supporting cells but not in hair cells. The induction of HSP70 prevented utricular hair cell death from cisplatin. Lovastatin appeared to protect the cochlea of cisplatin treated mice from ototoxicity by increasing the concentration of HSP60 and HSP32 mRNA in the cochlea [54]. The protective role of HSP32 (also known as heme oxygenase-1) against cisplatin ototoxicity was demonstrated in HEI-OC1 cells overexpressing the transcription factor, nuclear factor-erythroid factor 2-related factor 2 (Nrf2). This protection appeared to be mediated by PI3 kinase-AKT signaling that promoted heme oxygenase upregulation [55]. Nrf2 is a major regulator of cellular oxidative stress response and maintenance of redox homeostasis [56]. Nrf2 Activation has been shown to be otoprotective in HEI-OC1 cells in cisplatin-induced apoptosis [57]. The role of Nrf2/HO-1 pathway in hearing loss has been discussed in detail in several reviews [46,58,59,60,61,62].

Guinea pigs treated with cisplatin with or without pre-treatment with geranylgeranylacetone (GGA), inducer of HSPs [63], were compared for hearing loss and oxidative stress. Cisplatin treated animals had significantly greater shifts in ABR thresholds and greater NO and lipid peroxidation levels and had diminished expression of HSPs than those pretreated with GGA. The cochleae of guinea pigs treated GGA had reduced expression of NO and lipid peroxidation products and showed up-regulation of HSP27, HSP40 and HSP70. It appears that GGA protection against cisplatin ototoxicity is related to increased expression of these HSPs and that HSPs reduce oxidative stress in the cochlea caused by cisplatin [64].

## 6. G-Proteins

G-protein coupled receptors (GPCRs) are located in cell membranes and respond to various stimuli. Otoprotective GPCRs in the cochlea include A_1_ adenosine receptors (A_1_AR), cannabinoid receptor 2 (CB2) and sphingosine-1-phosphate receptor 2 (SIP2). A_1_AR have been demonstrated in the organ of Corti, stria vascularis and spiral ganglion cells [65]. In chinchilla, the A1AR agonist R-phenylisopropyladenosine (RPIA) activates these receptors and leads to increased activities of antioxidant enzymes, including GPx and SOD, in the cochlea [66]. Pretreatment with RPIA reduces cisplatin hearing loss by activation of A1AR causing decreased expression of *NOX3* and activation of STAT1 which reduced oxidative stress and inflammation [67]. CB2 receptors have been demonstrated in the rat organ of Corti, stria vascularis and spiral ganglion neurites [68]. Local application of JWH-015, a CB2 agonist, significantly reduced OHC death, ABR threshold shifts and synaptopathy. Blockade of CB2 receptors alone produces hearing loss, suggesting that there may be tonic activation of CB2 receptors under normal conditions [68]. A specific S1P2 receptor agonist, CYM-5478, reduced cisplatin ototoxicity in rats [69]. The use of agonists for cochlear-specific GCPRs may provide be an effective clinical approach to ameliorating cisplatin induced hearing loss [70]. Cisplatin was found to induce the expression of the *RGS17* gene leading to elevated levels of RGS17 protein. This contributed to hearing loss in rats. Conversely knockdown of *RGS17* ameliorated hearing loss induced by cisplatin. Regulators of G protein signaling (RGS) increases the GTPase activity of G proteins to promptly terminate the activation of GPCRs. RGS 17 appears to contribute to cisplatin ototoxicity by uncoupling protective GPCRs, such as CB2, from their normal protective actions. RGS17 inhibitors could provide novel agents for ameliorating cisplatin induced hearing loss [71] by extending the period of activation of GPCR activation.

## 7. Inflammation

Cisplatin induces the production of pro-inflammatory cytokines, including tumor necrosis factor-alpha (TNF-alpha), interleukin-1-β (IL-1β) and nuclear factor kappa B (NF-kB) in the cochlea [55]. Cisplatin induced activation of NF-kB increases the production of more pro-inflammatory cytokines, leading to activation of caspases 3 and 9, and increases in the expression of *inducible nitric oxide synthase (iNOS)*. This leads to the production of the free radical nitric oxide (NO) [72]. The oral administration of flunarizine greatly attenuated pro-inflammatory cytokine increases in the cochlea and serum and diminished the expression of *NF-kB* in the cochlea of cisplatin treated mice by activation of Nrf2/heme oxygenase-1 [55].

Cisplatin also induces the protein expression of signal transducer and activator of transcription-1 (STAT1) and down-regulates the expression of signal transducer and activator of transcription-3 (STAT3) in the cochlea [73]. ROS promote activation of STAT1 [74]. Activation of STAT1 in utricular hair cells in vitro promotes cell death [75]. Cisplatin increased the expression of *COX-2*, *iNOS* and *TNF-α* in rats. This increased inflammation in the cochlea leads to apoptosis of OHCs and loss of hearing [67,73]. STAT1 knockdown using siRNA in rats [67] or STAT1^−/−^ mice [76] were protected against cisplatin ototoxicity. 

STAT1 and STAT3 have opposing effects in the cochlea. STAT1 is pro-inflammatory and appears to mediate apoptosis in the cochlea resulting from cisplatin ototoxicity. STAT3 acts as a pro-survival molecule that helps to resolve inflammation [67,73]. Deletion of STAT3 sensitizes cells to oxidative stress [77]. STAT3 not only protects against oxidative stress, but it also up-regulates repair mechanisms for DNA damage [78]. Additional targets of STAT3 that promote survival are: the anti-apoptotic molecules Bcl-xL and Bcl-2 [79]. LMO4 targets STAT3 which confers pro-survival signals [80]. Cisplatin reduces the activity of LMO4 by nitration [28]. Animals deficient in LMO4 are more susceptible to cisplatin induced apoptosis in the cochlea and hearing loss [29]. The use of inhibitors for STAT1 to reduce cisplatin-induced inflammation and apoptosis may provide future treatments for cisplatin ototoxicity. In addition, the application of drugs to up-regulate *STAT3* expression in the cochlea could promote cell survival pathways could provide novel therapeutic approaches to ameliorate cisplatin ototoxicity.

## 8. Current Treatments

A wide variety of putative protective agents against cisplatin ototoxicity have been studied [70]. Antioxidant drugs have demonstrated protection against ototoxicity in animals treated with cisplatin [81]. Sodium thiosulfate reduced hearing loss in animal models with systemic administration. Unfortunately, systemically administered sodium thiosulfate reacts with cisplatin and reduces its anti-tumor activity [82]. Intratympanic injection avoids this problem [83]. Delayed administration of sodium thiosulfate has shown promise in clinical trials. This protective agent was administered intravenously in children with cancer 6 h after receiving cisplatin. This treatment showed significant reduction of the probability of hearing loss; however, patients with disseminated cancer were found to have a reduced survival rate [84]. In children with localized hepatoblastoma, the delayed administration of sodium thiosulfate 6 h after cisplatin reduced hearing loss without any discernible difference in survival [85]. A phase 3 clinical trial utilized cisplatin injection into the nutrient artery supplying the tumor in patients with locally advanced head and neck cancer. Sodium thiosulfate was injected in a different route (intravenously) at the same time. This group showed a 10% decrease in hearing loss at the higher frequencies [86]. 

Another thiol compound, N-acetylcysteine was effective in ameliorating hearing loss and cochlear damage in rats [87,88]. D-methionine, a sulfur-containing amino acid, was shown to be effective in reducing cisplatin ototoxicity when administered systemically [89] or locally [83,90]. The protection mediated by D-methionine appeared to be due to increased expression of antioxidant enzymes in the cochlea [91]. Other sulfur-containing antioxidants which showed efficacy against cisplatin-induced hearing loss in rats include lipoic acid, diethyldithiocarbamate and 4-methylthiobenzoic acid [92]. Amifostine is an organic thiophosphate cytoprotective agent. High doses of amifostine provided otoprotection in hamsters but its use was associated with neurotoxicity [93]. The selenium compound, ebselen is a glutathione peroxidase mimetic. Ebselen protected against cisplatin ototoxicity and nephrotoxicity in rats when combined with allopurinol [25].

Intratympanic administration of RPIA, an adenosine A1AR receptor agonist, increased the production of antioxidant enzymes (GSH-Px) and SOD in the cochlea [66] and protected OHCs from damage and prevented hearing loss [94] by reducing the expression of NOX3 ROS production and STAT1 mediated inflammation [67]. Cisplatin ototoxicity was also ameliorated by treatment with the adenosine amine congener (ADAC), acting via the A1AR [95].

Activation of CB2 receptors in the cochlea by intratympanic administration of JWH-015, significantly protected against cisplatin induced hearing loss and OHC cell death and synaptopathy. This protective effect appeared to be mediated, in part, by inhibition of STAT1 [68].

It is critically important that any protective agent given to protect against cisplatin ototoxicity not interfere with the anti-tumor efficacy of cisplatin. A number of animal studies tested putative protective agents by intratympanic injection. Drugs such as steroids, etanercept, d- and l-methionine, pifithrin-alpha, adenosine agonists, melatonin, kenpaullone (a cyclin dependent kinase 2 [CDK2] inhibitor) have been reported to show efficacy against cisplatin ototoxicity in rodent models (reviewed in [96]. The administration of protective agents by this route would likely avoid inactivation of cisplatin in the systemic circulation. 

It is encouraging to note that several studies utilizing systemic administration of protective agents have found amelioration of cisplatin ototoxicity in experimental animals without interference with its chemotherapeutic efficacy. Nude rats bearing HT-29 tumors received cisplatin in combination with a Src-protein kinase inhibitor. This co-treatment protected against cisplatin ototoxicity without compromising the efficacy of cisplatin in tumor response [97]. Oral β-lapachone ameliorated cisplatin ototoxicity. Cisplatin administration depleted intracellular NAD^+^ and SIRT1 levels in the cochlea, but these levels were augmented by β-lapachone. β-lapachone decreased cochlear ROS production and DNA damage induced by cisplatin. β-lapachone was also found to reduce the production of pro-inflammatory cytokines in the cochlea by inhibiting NF-kB acetylation [98]. Cisplatin greatly increased the sensitivity of β-lapachone in suppressing the growth of leg tumors in mice [99]. Pifithrin-alpha is a p53 inhibitor that protected mice from ototoxicity without interference with cisplatin chemotherapeutic effectiveness and even sensitized p53-mutant tumors to cisplatin [13]. Epigallocatechin-3-gallate (EGCG), the polyphenol extract from green tea, was administered by oral gavage to cisplatin treated SCID mice bearing human head and neck squamous cell carcinoma cells. There was no evidence of interference of cisplatin anti-tumor efficacy [100]. Similar results were shown in tumor-bearing SCID mice treated with cisplatin and oral capsaicin. Tumor killing by cisplatin was not impaired by capsaicin co-treatment [73]. Honokiol, a polyphenol derived from *Magnolia officinalis*, ameliorated cisplatin ototoxicity in mice bearing mammary tumors were treated with cisplatin. Honokiol, did not interfere with the ability of cisplatin to effectively shrink the tumors and acted synergistically with cisplatin to reduce tumor size [101].

Intraperitoneal lovastatin injections prior to and after cisplatin administration protected against ABR threshold shifts and reduced OHC loss in mice. Lovastatin significantly increased the levels of the mRNA for heat shock proteins (HSP60 and HSP32/Hmox1) in the cochlea [54]. A very significant clinical trial compared head and neck cancer patients treated with cisplatin with or without concurrent atorvastatin. Those patients receiving atorvastatin had a significantly lower incidence of hearing loss compared to patients not receiving atorvastatin. Cancer patients on atorvastatin were found to be 53% less likely to suffer hearing loss compared to patients not taking a statin drug. Most important was the finding that 3 year survival rates were similar in both groups of patients [102].

## 9. Future Directions

There are several potential treatments to explore in the future for prevention of cisplatin ototoxicity. Animal studies which have shown drugs that protect against cisplatin ototoxicity without interfering with cisplatin efficacy should be considered for clinical trials. These include β-lapachone, honokiol, pifithrin alpha, EGCG, capsaicin, and a Src-protein kinase inhibitor.

Pediatric oncologists recommend trials of systemic rather than intratympanic route of treatment with protective agents against cisplatin [103]. A strong recommendation for continued use of systemic sodium thiosulfate in children with nonmetastatic hepatoblastoma was made. Further research to determine the safety of sodium thiosulfate in patients with metastatic cancer was encouraged [104]. For adults receiving cisplatin, the transtympanic approach appears to be attractive. However optimal dose, number of injections and clarifying what protectant is the safest and most effective would be important factors for consideration [105]. A novel formulation of sodium thiosulfate appears to be well tolerated and safe. Further studies are needed to determine its efficacy against cisplatin ototoxicity [106,107].

ORC-13661 is a novel high-affinity permeant blocker of the mechanoelectrical transducer (MET) channel in OHCs. In mouse cochlear cultures it provided significant protection of hair cells against cisplatin. It appears to be well tolerated in rats following oral administration [18]. Future studies should investigate whether this drug reduces cisplatin ototoxicity in mammalian models as it has been shown to protect rats against aminoglycoside ototoxicity [18].

Statins do not appear to interfere with cisplatin’s anti-tumor effects in patients [102]. Additional clinical trials with statins should be carried out. β-lapachone appears to be safe and well-tolerated in early clinical trials of oral administration in healthy human volunteers [108]. It seems to be a promising agent for protecting against cisplatin ototoxicity and is worth considering for clinical trials.

Exosomes are membrane-bound nanovesicles that can be released from various cells in the body including supporting cells in the cochlea. They can provide HSPs to protect against cisplatin ototoxicity [109]. They could be used as nanocarriers to deliver drug treatments [109] or stem cells [110] to treat hearing loss from cisplatin chemotherapy. These nanovesicles appear to have great potential to protect against cisplatin induced hearing loss. 

## 10. Conclusions

Cisplatin chemotherapy is associated with a very high incidence of irreversible sensorineural hearing loss. The drug gains access to the cochlea by several transport mechanisms, including the MET channels, which could serve as targets for treating ototoxicity as shown in Figure 1 and tabulated in Table 1. The generation of ROS and inflammation by cisplatin appear to be integral to mediating cisplatin ototoxicity. There are several novel agents that can protect against cisplatin ototoxicity. Some of these drugs have shown excellent protection of hearing without interfering with the therapeutic effect of cisplatin. The future looks bright and future clinical trials could bring much needed protection against cisplatin ototoxicity.

## Figures and Tables

**Figure 1 antioxidants-10-01919-f001:**
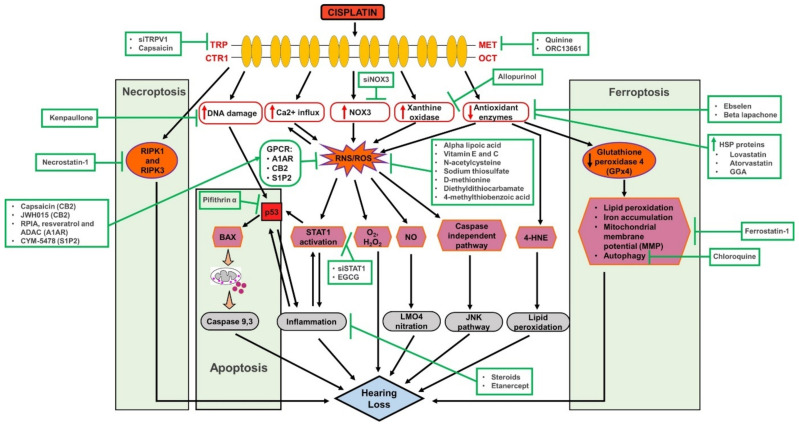
Comprehensive figure of cisplatin ototoxicity and potential targets for otoprotection. Cisplatin entry through several membrane sites including MET (mechanoelectrical transduction), TRP (transient receptor potential) and OCT (organic cation transporters) channels. Several mechanisms of cisplatin ototoxicity are listed, namely the generation of ROS via NOX3, leading to the activation of lipid peroxidation, DNA damage, inflammation and cell death pathways. Several potential targets for otoprotection include inhibition of drug entry (via MET channels), scavenging of ROS (using antioxidants), inhibition of inflammation (using anti-inflammatory drugs) and inhibition of apoptosis (using p53 inhibitors).

**Table 1 antioxidants-10-01919-t001:** Potential drugs and their targets for the treatment of cisplatin ototoxicity.

Experimental Drug		Mechanism of Action	Animal Model	Route of Administration	Reference
1	Amifostine	Free radical scavenger	Hamster	Intraperitoneal	[93]
2	Atorvastatin	Commonly used drug for management of hypercholesterolemia by inhibition of 3-hydroxy-3-methylglutaryl-CoA (HMG-CoA) enzyme	Human	Oral	[102]
3	β-Lapachone (NAD+)	Anti-oxidant	Rat	Oral	[98]
4	Capsaicin	TRPV1 agonist that desensitizes CB2R agonist	Rat	Oral	[73]
5	CYM-5478	SIP-2 receptor agonist	Rat	Intraperitoneal	[69]
6	D-Methionine (D-Met)	Anti-oxidant molecule	Rat	Intraperitoneal	[89]
7	Ebselen	Glutathione peroxidase mimetic	Rat	Oral	[25,111]
8	EGCG	STAT1 inhibition	Rat	Oral	[100]
9	Flunarizine	Nrf2 activation	Mice	Oral	[55]
10	GGA	Inducer of HSPs	Guinea pig	Systemic	[64]
11	Honokiol	Anti-oxidant	Mice	Intraperitoneal	[101]
12	JWH-015, (2-methyl-1-propyl-1H-indol-3-yl)-1-naphthalenylmethanone)	CB2R agonist	Rat	Intratympanic	[68]
13	N-acetylcysteine	Anti-oxidant	Rat	Intravenous	[87]
14	Lovastatin	Commonly used drug for management of hypercholesterolemia by inhibition of 3-hydroxy-3-methylglutaryl-CoA (HMG-CoA) enzyme	Mice	Oral Gavage	[54]
15	Pifithrin-α	p53 inhibitor	Mice	Intraperitoneal	[13]
16	RPIA, ADAC	A1AR agonist	Rat	Intratympanic	[67,95]
17	STAT1 siRNA	Anti-inflammatory	Rat	Intratympanic	[112]
18	TRPV1 siRNA	Anti-inflammatory	Rat	Intratympanic	[40]
19	Sodium thiosulphate	Anti-oxidant	Guinea pig Humans	Round window administration	[83] [104]

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
