# Peer review of "Oxidative Stress and Inflammation Caused by Cisplatin Ototoxicity"

_antioxidants, 2021, doi:10.3390/antiox10121919_

Round 1
Reviewer 1 Report
Great review. I recommend a few modification.
Please describe the abbreviations when you use them for the first time
Rephrase between row 71 and 85 its hard to read!
Article it's difficult to read!
I think that you need three short conclusions! My recommendation is to focus on 3 short conclusion.
Thank you again for the opportunity to review this interesting manuscript.
Author Response
We have made all the requested revisions. Thank you.

Reviewer 2 Report
The manuscript by Ramkumar et al. reviews what is known about the different pathways of ototoxicity of cisplatin and the compounds ameliorating this toxicity, with suggestions for future clinical trials.
MAJOR CONCERNS
(1) The review is quite thorough. However, I miss an integration of data. Clearly, a figure summing up how all damage pathways add up and may interact is essential for the reader not to lose focus.
(2) Care should be taken to clearly explain the different roles in toxicity of hair cells and supporting cells in the organ of Corti, of spiral ganglion neurons and of the stria vascularis. If these interactions are unknown, it would be adequate to point it ans suggest it as a future direction for research.
(3) Exosomes are mentioned almost as an afterthought. I miss a short explanation on how exosomes would be used to alleviate cisplatin ototoxicity, including delivery routes. Would they be used in combination with any of the previously discussed drugs or compunds?
MINOR CONCERNS
Sentence in lines 87-89, with two "in orders" in tandem seems awkward. Could you please rephrase it?
Author Response

(The authors gave the same response as above.)
